# Enhancement of Growth and Secondary Metabolites by the Combined Treatment of Trace Elements and Hydrogen Water in Wheat Sprouts

**DOI:** 10.3390/ijms242316742

**Published:** 2023-11-25

**Authors:** Muniba Kousar, Yu Rim Kim, Ji Yeon Kim, Joonho Park

**Affiliations:** 1Department of Fine Chemistry, Seoul National University of Science and Technology, 232-Gongneung-ro, Nowon-gu, Seoul 01811, Republic of Korea; 2Department of Food Science and Technology, Seoul National University of Science and Technology, 232-Gongneung-ro, Nowon-gu, Seoul 01811, Republic of Korea; 3Center for Functional Biomaterials, Seoul National University of Science and Technology, 232-Gongneung-ro, Nowon-gu, Seoul 01811, Republic of Korea

**Keywords:** wheat leaves, hydrogen water, trace elements, secondary metabolites, response surface methodology

## Abstract

This study aimed to evaluate the response of *Triticum aestivum* to hydrogen water (HW) and trace elements treated with HW. A pot experiment was conducted to assess the growth indices, secondary metabolites, and antioxidant levels. The response surface methodology (RSM) approach was used to ascertain the concentrations and significant interaction between treatments. The outcomes demonstrated that the combined treatment of Se acid and Mo oxide exhibited a notable positive effect on the growth and secondary metabolites, when treated with HW as compared to distilled water (DW). Notably, the interaction between these two treatments is significant, and the higher response was observed at the optimal concentration of 0.000005% for Se acid and 0.06% for Mo oxide. Additionally, an in vitro experiment revealed that the mixture treatment inhibits the accumulation of lipids in HepG2 hepatocytes cells. Moreover, metabolic analysis revealed that upregulated metabolites are linked to the inhibition of lipid accumulation. In addition, the analysis emphasizes that the continued benefits of higher plants as a renewable supply for chemicals compounds, especially therapeutic agents, are being expanded and amplified by these state-of-the-art technologies.

## 1. Introduction

*Triticum aestivum* L. is arguably one of the most dominant and most cultivated crops around the world. It is considered the third-largest cereal crop, with more than 600 billion tones harvested annually [1]. Global food security is linked to the production and consumption of wheat [2], as this crop provides 19% and 21% of the world’s daily calories and protein requirements, respectively, which are crucial for human needs [3].

Photocatalytic water splitting is used to produce HW, on account of several advantages of this process, such as its excellent photochemical stability, uncomplicated procedures, non-toxicity, and low cost [4,5]. According to recent research, hydrogen molecules may improve the capacity of tissues to respond to stress [6]. The most significant physiological regulatory component over the past ten years has been identified as hydrogen molecules, which have outstanding anti-oxidative and anti-inflammatory capabilities and can protect against both inflammation and oxidative damage [7,8]. Hydrogen can be inhaled or simply dissolved in water to produce hydrogen water (HW). Notably, one study contends that the consumption of HW can significantly enhance the lipid profile of clinical populations [9]. The ability of hydrogen to neutralize hydroxyl radicals and peroxynitrite anions makes it a potent antioxidant [10]. It has been suggested that agriculture techniques should use hydrogen as a treatment for plants [11]. According to the study, hydrogen-rich water enhanced many phytohormones and extended the shelf life of postharvest okras, showing that it has the potential to improve fruit quality [12]. One study found that hydrogen plays a role in enhancing the activity of antioxidant enzymes and the transcription of related genes, which can promote the growth of mung bean and rice seedlings and roots [13]. Additionally, the HW treatment significantly decreased the tendency of litchi to oxidize by raising its antioxidant capacity [14]. Furthermore, HW treatment can both prevent and delay the ripening of banana fruit as well as the degradation of the starch and cell wall [15]. There is growing evidence that hydrogen injection alters cell antioxidant capacity [16,17], despite the fact that a direct relationship with the molecular target of hydrogen has yet to be identified. According to reports, heme oxygenase-1 mediates the effects of hydrogen and is responsible for some of the reactions reported in plants [18,19], as well as in animals [20,21]. In fungi, a selenium-based enzyme called glutathione peroxidase has been demonstrated to mediate HW effects [22]; however, the precise mechanism of how hydrogen interacts is still unknown.

Trace elements have been studied extensively due to their importance in relation to plant growth and animal nutrition [23]. Elements that are present at low concentrations are termed as trace elements. Some trace elements are termed micronutrients and play a key role in plant growth. Examples include molybdenum (Mo), zinc (Zn), and copper (Cu) [24]; however, some are not essential for plant growth such as selenium (Se) and cobalt (Co) but play a key role in the growth of plants and are thus beneficial for humans. Plants play a vital role in the cycling of all trace elements in the agricultural ecosystem. However, the role of trace elements in plants remains a topic of much research. In this study, two forms of Molybdenum (Mo (vi) oxide and Mo chloride) and Se acid are selected as the working treatments. Selenium is regarded as non-essential, however, because of its role as an antioxidant, and in the control of ROS, it can be a beneficial element for plant growth [25]. Selenium has been observed to increase the total flavonoid and phenolic content of wheatgrass sprouted from selenium-rich wheat grains, demonstrating that selenium may influence the antioxidant content of wheat plants [26]. The most crucial element for higher plants is molybdenum [27]. It is involved in metabolism and regulates sulfur, carbon, and nitrogen through a variety of molybdenum cofactors [28]. An earlier study found that Mo improved micro- and macronutrient allocations, which, in turn, boosted the uptake of micro and macronutrients in wheat [29]. The secondary metabolites of plants are eccentric sources of compounds used in the pharmaceutical industry, as well as for food additives, cosmetics, flavors, nutritional products, and dietary supplements [30]. In addition, secondary metabolites aid in environmental communication and plant defense [31].

Metabolomics is the thorough analysis of metabolites in biological systems, and it was developed rapidly in the mid-1990s [32,33]. It is an effective tool for comprehending how plants regulate their primary and secondary metabolites and their metabolic pathways [34]. The metabolomics of the plant secondary metabolism has been established for both functional genomics and the investigation of physiological processes using multi-omics techniques [35]. The widespread use of mass spectrometry for metabolite profiling has made the separation and identification of metabolites and intermediates in plant tissue possible [36]. Mass spectrometry was used to identify the changes in metabolites under the combined treatment of trace elements and hydrogen water. 

This study represents experiments indicating that combining various antioxidants strategies will be more beneficial than only enhancing the antioxidant capacity by a single treatment. The objective of this study is to promote specific secondary metabolites, which can inhibit lipid accumulation, by combining HW and trace elements. The interaction mechanism presumably involves hydrogen’s capacity to maximize the usage of trace elements inside biological processes, thus increasing cellular functioning and overall treatment efficacy. Thus, to investigate the optimal concentration of trace elements treated with HW, the total flavonoid content, total polyphenolic content, and antioxidant content was determined. Moreover, RSM was used to determine significant interaction between HW -treated trace elements by using the optimal concentration. The metabolic profile of wheat leaves was identified by using chromatography to determine the effect of hydrogen-water-treated trace elements. The potential of the mixture treatment extract to lower fatty lipid accumulation in HepG2 cells was evaluated. This study provides a new strategy to figure out the effect of HW-treated trace elements on crops.

## 2. Results

### 2.1. Effects of the Non-Optimal Concentrations of Trace Elements Treated with HW on the Biomass Parameters of Wheat Leaves

Wheat growth outcomes were determined using different concentrations of each element treated with HW and distilled water. After various trials, the initial results indicated that a concentration of Mo oxide of 0.01%, Se acid of 0.000001%, and Mo chloride of 0.001% can increase the growth of wheat leaves.

Table 1 shows that HW can significantly increase the biomass characteristics of wheat leaves compared to the control (ANOVA; *p* < 0.01). In contrast to their treatment with distilled water, Mo oxide (0.01%) and Se acid (0.000001%) treated with HW can dramatically accelerate growth. When compared to the distilled water treatment, the Mo oxide (HW) showed a 27% increase in area, while the Se acid (HW) showed a 38% increase in the area. However, the area of wheat leaves was reduced when Mo chloride (0.001%) was treated with HW (almost 27%), but there was no discernible difference in the length or weight of the leaves when Mo chloride was treated with distilled water.

### 2.2. Effects of the Non-Optimal Concentrations of Trace Elements Treated with HW on the Total Flavonoids and Total Polyphenol Content of Wheat Leaves

The HW treatment, as compared to the distilled water treatment, significantly increased the rate of secondary metabolites. As compared to Mo oxide 0.01% treated with DW, the Mo oxide treated with HW increased the total flavonoid and total polyphenol contents by almost 40% and 30%, respectively (ANOVA; *p* < 0.05), likely because molybdenum plays a key role in enhancing phytochemical production and thus can increase secondary metabolites production in wheat leaves. Meanwhile, the Se acid-HW 0.000001% treatment increased the total flavonoid content by 35% and the total polyphenol content by 55% (ANOVA; *p* < 0.05), whereas exposure to Mo chloride-DW treatment resulted in a significant decrease (almost 24%) in total polyphenol content as compared to control. However, the Mo chloride-HW treatment increased the total polyphenol content by 15% as compared to the Mo chloride-DW treatment. In contrast, the Mo chloride-HW treatment increased the total flavonoid content by 27% as compared to the control, as shown in Figure 1.

### 2.3. Effects of the Non-Optimal Concentrations of Trace Elements Treated with HW on the Antioxidant Content of Wheat Leaves

The DPPH (2,2-diphenyl-1-picrylhydrazyl) assay is based on the assessment of the scavenging capacity of antioxidants. In this study, the radical scavenging activities of the Mo oxide and Se acid-HW treatments are relatively higher in comparison with the control and compared to when they were treated with DW (*p* < 0.05). They increased the antioxidant capacity by 38% and 35%, accordingly, as compared to control, as illustrated in Figure 2. However, the Mo oxide-DW and Se acid-DW treatments increased the antioxidant content by 23% and 32%, respectively, as compared to the control group. On the other hand, no significant difference was observed following the application of Mo chloride 0.001% treated with HW and distilled water, though a slight increase was noticed as compared to the control.

### 2.4. Optimizing the Concentration with Response Surface Methodology

Response surface methodology (RSM) is a statistical, theoretical, and mathematical model-building technique used to optimize the level of independent variables. The experimental results of the HW-treated Mo oxide and Se acid were utilized to optimize the concentration of each element presented in Table 2 for each of the two compounds.

Previously, we determined the three different ranges of Mo oxide and Se acid to obtain the response value for RSM. In order to obtain a clarified view, the 2D surface plot and 3D contour plot were illustrated by showing the total flavonoid response level. The results indicate that the RSM model is significant, as made obvious by the circular or elliptical appearance of the contour plot in Figure 3a,b. Moreover, our data revealed that both Mo oxide-HW and Se acid-HW positively affected the total flavonoid level of wheat extract. The darker green region in Figure 3b depicts the greater rate of flavonoid content (higher than 180 mg/mL) as a result of different ranges of Mo oxide and Se acid concentration.

The factorial plot in Figure 3c represents the range of the optimal concentrations of these treatments. It indicates that the Mo oxide concentration range is between 0.01 and 0.06% while the range for Se acid is 0.000001–0.000006%, demonstrating that the maximum response rate of the total flavonoid content is 150 to >180 mg/mL. Importantly, however, a value higher or lower than these levels will have a negative impact on the overall flavonoid content. The optimal concentration range of these trace elements is identical to the total polyphenol and antioxidant content represented in Appendix A.

The interaction plot in Figure 3d demonstrates that the interaction between the Se acid and Mo oxide treated with HW is significant. However, it is crystal clear that the interaction at the concentration of 0.06% Mo oxide and 0.000005% of Se acid is significant. From our previous experiments with the initial concentration that we tested, the findings indicated that the concentrations of 0.01% for Mo oxide and 0.000001% for Se acid can increase the secondary metabolites and antioxidants of wheat leaves. However, the RSM data provide the optimal ranges and optimal concentration for the interaction of these treatments from which the maximum response of secondary metabolites and antioxidant content can be obtained. RSM is mostly used to cut down the number of experiments to obtain the optimization condition more rapidly.

### 2.5. Wheat Cultivation, Extraction, and Determination of Secondary Metabolites under Optimal Conditions

The results demonstrated that the optimal concentration of trace elements in a treatment with HW can increase the rate of secondary metabolites. Mo oxide and Se acid under the optimal concentration significantly increased the total flavonoid content by 22% and 12.5%, respectively, as compared to the non-optimal concentration Figure 4a.

Meanwhile, as shown in Figure 4b,c, the optimal concentrations of Mo oxide-HW resulted in a 13% increase in total polyphenol levels and a 43% increase in antioxidant levels when compared to the control group. Similarly, appropriate Se acid-HW concentrations resulted in a 6% increase in total polyphenol levels and a 33% increase in antioxidant levels. On the other hand, the mixture of both trace elements treated with HW at the optimal concentration increased the secondary metabolites to a greater level. The rate of total flavonoid and total polyphenol increased by mixture treatment is 42.4% and 19.5%, respectively, as compared to DW, as well the antioxidant content is also increased by 47% by mixture treatment.

As the mixture of both treatments showed significant enhancement in the secondary metabolites and antioxidant content, we selected this treatment for comparison with the control treatment for further analysis.

### 2.6. Effect of Mixture Treatment on the Metabolic Profile of Wheat Leaves

To depict the general classifying information, an unsupervised multivariate technique based on the similarities between the metabolic profiles, which was accomplished with mean linkage and Pearson’s correlation analysis, was used. The chromatograms generated by using a single-ion monitoring system are in the positive and negative ion mode. A total of 191 peaks were observed in both negative and positive mode, based on the retention time and mass spectra analysis. Of 191 observed peaks, 47 metabolites peaks were identified.

In the present study, the heatmap dendrogram shows that the identified metabolites play pivotal roles in crucial metabolic pathways, including organic acids, amino acids, and fatty acids. Furthermore, after exposure to the mixture (Se acid and Mo oxide treated with HW) treatment, a notable shift in metabolite levels was observed in wheat leaves, with approximately 61% showing elevation and around 19% displaying downregulation.

Using multivariate component analysis, the most significantly upregulated metabolites are vitexin derivative metabolites, and saponarin, threonylleucine, and isoleucine were observed (Figure 5a). Additionally, by combining the *p*-value and pathway impact values, we identified some significant metabolic pathways in Figure 5b. Our pathway analysis demonstrated that the significantly affected metabolic pathways are linoleic acid metabolism, glycerophospholipid metabolism, citrate (TCA) cycle, glyoxylate and dicarboxylate metabolism, glycerophospholipid metabolism, phenylalanine metabolism, and biosynthesis of secondary metabolites. 

### 2.7. Effects of the Mixture Treatment on Lipid Accumulation and the Triglyceride Content

The cytotoxicity of the experimental concentrations was initially analyzed before treating the HepG2 cells with a mixture of treatment extracts. The results indicated that at 25% concentration of solvent ethanol (80.6%), the control sample (DW, 91.64%) and sample (mixture, 81.09%) showed that cell viability exceeds 80%. The results shown in Appendix A allowed us to set the experimental concentration to 25%.

The cells were initially treated with a mixture extract for 24 h to assess lipid accumulation using oil red O staining (ORO). The results illustrated the capability of OA (oleic acid) to induce lipid accumulation. No steatosis was observed using ORO staining in the control group; however, after OA treatment, the accumulation of lipid droplets in the cytoplasm of HepG2 cells was observed. The morphology of the stained cells with lipid accumulation is shown in Figure 6a. The presence of distinct red lipid droplets surrounding the cells in the model group indicated a successful induction group. The accumulation of lipid droplets amounted to 231.52% in the induction group (OA), 226.46% for SP-CON, and 212.42% in the SP group. As compared to the OA-induction group, the SP (mixture) treatment showed reduced lipid accumulation and significantly enervated lipid fusion in HepG2 cells after 24 h, as depicted in Figure 6b.

To determine the triglyceride (TG) accumulation, we measured the TG level in HepG2 cells. The results demonstrated that the concentrations of triglycerides were 57.11 mg/dL in the induction group (OA), 52.61 mg/dL in the SP-CON group, and 39.31 mg/dL in the SP group. Overall, the SP group had better triglyceride lowering effect as compared to the OA-induction group, as shown in Figure 7.

## 3. Discussion

This research provides new knowledge on how trace elements, when combined with HW, impact the production of secondary bioactive metabolites, which can be utilized to produce wheat products that encompass higher levels of targeted bioactive compounds.

Plants also contain some antioxidants and secondary metabolites with low molecular weight, such as phenols, flavonoids, tocopherols, and anthocyanins [37,38]. The hydroxyl groups present in polyphenols and flavonoids exert an antioxidant effect by binding to reactive oxygen species (ROS) [39].

Hydrogen gas (H_2_) plays an important role in the resistance of plants to stress responses [40]; however, the potential effects of hydrogen-enriched water and the mechanism behind the HW action on plant development and the production of secondary metabolites is still unclear. This study reveals the role of HW in enhancing secondary metabolites and antioxidant content of wheat leaves. 

The level of active compounds present in plants depends on their environment throughout growth. This study looked at several circumstances to increase wheat leaf secondary metabolites. The germination condition includes three trace elements, subjected to HW treatment in different concentrations. It is evaluated to determine which of these trace elements may boost the total content of secondary metabolites in wheat leaves. 

The response surface methodology was used to obtain the content of total polyphenol, flavonoid, and antioxidant content to ascertain the optimal range concentrations of these trace elements. This study also reveals the significant interaction between these two trace elements and the effects on the secondary metabolites of wheat leaves identified using the response surface methodology. In one study, the RSM model was also shown to be suitable to predict the optimization of antibiotic production conditions [41]. In this study, we observed the optimal ranges for Mo oxide and Se acid to be 0.01–0.06% and 0.000001% to 0.000006%, respectively. However, a significant interaction was noticed at the concentrations of 0.06% for Mo oxide and 0.000005% for Se acid. This optimal concentration was chosen to test the growth and secondary metabolites of wheat leaves.

Mo is a necessary trace element for plants. Overall, an organism contains several Mo-dependent enzymes; however, only five of them have been observed in plants [42]. These Mo enzymes play a role in the metabolism of sulfur, purine, nitrogen, and phytohormones in plants. The secondary metabolic process of glycyrrhizin acid in Glycyrrhiza uralensis Fisch has been shown in the past to be influenced by Mo [43]. In our study, it was discovered that Mo oxide-HW can increase the content of secondary metabolites. The most significant increase was observed in the antioxidant content of wheat leaves following Mo oxide-HW treatment. This increase is likely caused by the fact that Mo must first complex with pterin compounds, a special protein known as molybdoprotein, which creates a molybdenum cofactor (Moco) for biological activity and is involved in the secondary metabolites of plants [44]. In general, only Mo oxide and Se acid treatments have been observed to increase the physiological growth and secondary metabolites of wheat leaves. However, the mixture of Mo oxide and Se acid treated with HW has been found to increase the overall secondary metabolites at a significant ratio.

Comparing the levels of total polyphenols, total flavonoids, and antioxidant content after RSM demonstrated that the rate of overall secondary metabolites is increased by the optimal concentration of these trace elements. Moreover, the verification experiment confirmed that the interaction between two treatments is significant. The RSM approach is becoming more popular to extract bioactive compounds because of its ability to determine how various independent variables can interact with each other. 

An essential metabolic pathway for the metabolism of biomolecules is the citric acid cycle, also known as the Kreb cycle. It plays a key role in the biosynthesis of various secondary metabolites such as amino acids, purines, flavonoids, and fatty acids by providing a crucial precursor, in addition to its primary function in energy production [45]. The expression of particular genes involved in secondary metabolites production can be influenced by the level of TCA cycle metabolites, such as acetyl-CoA, succinyl-CoA, and OAA [46]. According to a study, there was a strong correlation between the glyoxylate and dicarboxylate metabolism of two pairs of genotyped eggplants that had different nitrogen usage efficiency in secondary metabolite production [47]. Our study revealed that the effect of mixture treatment altered various metabolic pathways such as the citrate cycle, glyoxylate and dicarboxylate metabolism, and pyruvate metabolism. These pathways are linked to the synthesis of different secondary metabolites, which is consistent with a study [48]. Moreover, our metabolic study revealed the enhancement in various metabolites after the mixture treatment, which may be linked with the inhibition of lipid accumulation according to the previous reports. Thus, an in vitro experiment was carried out with the mixture treatment grown under optimal conditions. 

The in vitro hepatocyte model was established using HepG2 cells [49,50]. In the case of non-alcoholic fatty liver diseases (NAFLD), we treated HepG2 cells with FFAs to notice the deviations in triglyceride accumulation (TG) [51]. The accumulation of TG in the liver is because of the increased level of TG and can be supplied by circulating FFAs in the blood [52]. Our results revealed that the accumulation of lipids can be inhibited by the mixture treatment as compared to the induction group. This may be possible because the various metabolites affected by the mixture treatment, as determined via metabolic analysis, have a beneficial role in controlling the accumulation of lipid by reducing the lipid level. For example, vitexin derivatives have a role in controlling the accumulation of fat in blood by upregulating the AMPKα signaling pathway [53]. Moreover, one study revealed that vitexin prevents non-alcoholic fatty liver disease (NAFLD) by activating AMPK, which decreases lipogenesis while increasing lipolysis and fatty acid oxidation [54]. The isoleucine upregulated by our treatment also seems to enhance the synthesis of unsaturated fatty acid and accumulation of fat in muscles of finishing pigs (which refer to pigs that have completed their growth cycle and are ready for market) [55]. Because of saponarin’s anti-inflammatory, anti-cancer, antioxidant properties, it has been shown to have potential therapeutic effects [56]. A study has shown that except for the 8 μM saponarin concentration of barley extract, all other saponarin concentrations can inhibit the accumulation of lipids [57]. Our study revealed that all of these metabolites are significantly increased by the combined treatment of trace elements and hydrogen water.

Overall, combining trace elements with HW under optimal concentrations can elevate the biosynthesis of secondary metabolites as well as inhibit lipid accumulation. We investigated the saponarin content of wheat leaves, opening the path for future research. Our long-term goal is to increase saponarin levels in wheat leaves using diverse methods, making wheat more appropriate for therapeutic purposes. 

## 4. Materials and Methods

### 4.1. Chemicals and Reagents

For the physiological growth experiment, all tested trace elements, namely Mo (VI) oxide, Mo (v) chloride, Selenous acid, and materials used for the determination of secondary metabolites and antioxidants content, namely Folin–Ciocalteu’s reagent, gallic acid (3,4,5-trihydroxybenzoic acid), catechin, and DPPH (2,2-diphenyl-1-picrylhydrazyl) reagent (2,2-diphenyl-1-picrylhydrazyl), were purchased from Sigma-Aldrich (St. Louis, MO, USA). For cell lining experiments, oleic acid (OA), bovine serum albumin (BSA), and Thiazolyl Blue Tetrazolium Bromide (MTT) were also purchased from Sigma-Aldrich, and DMEM (Dulbecco’s modified eagle’s medium), FBS (fetal bovine serum), and the penicillin–streptomycin mixture were obtained from Biowest (Nuaille’, Cholet, France).

### 4.2. Experiment Design

Seeds of wheat (*Triticum aestivum* L., Keumgang variety) were purchased from Danong. Co., Ltd. (Seoul, Republic of Korea), and were kept at 2–8 °C until use. Stock solutions of each element were prepared and diluted using HW and distilled water. We prepared the hydrogen water in the lab using a hydrogen water generator (SOLCO, P Gyeonggi-do, Republic of Korea). We poured 1000 mL of distilled water into this generator and let the hydrogen dissolve into the water for 30 min. Basically, the electrolysis system with 6 layers of titanium–platinum allows water (H_2_O) to be quickly and securely divided into hydrogen (H_2_) and oxygen (O_2_), allowing hydrogen to rapidly spread out into the water molecules to be dissolved. Once the hydrogen had adequately dissolved, we collected the hydrogen water for experimental use. For the pot experiment, three concentration ranges for optimization were selected for each trace element (Mo oxide, Mo chloride 0.1–0.001%, and Se acid 0.00001–0.000001%). The exposure solution of each treatment was prepared via stock solution. The wheat seeds were washed three times to remove possible impurities. Afterward, the 25 uniform seeds were transferred to the soil (almost 25 seeds in one pot). The pot used in this experiment was 15 cm high with a diameter of 12.5 cm at the bottom and 18 cm at the top. The plants were grown under white light (light intensity was set at 200 μmol m^−2^ s^−1^), and the temperature was set to 25 ± 2 °C. Wheat seedlings were exposed to HW-treated trace elements and DW-treated trace elements. DW was set as the control group. Each treatment’s test solution was given after a day. Three replicates of each treatment were carried out. After the seventh day of germination (i.e., on the 8th day), the wheat leaves were harvested. After measuring the growth parameters, the wheat samples were stored at −80 °C for further analysis. The growth indices, which include the area and length, were analyzed using ImageJ 1.51K software (Java 1.6.0_24 (64-bit), San Diego, CA, USA), and the weights were measured with a Secura513 balance (Sartorius, Seongnam, Republic of Korea).

### 4.3. Extraction Procedure of Wheat Leaves

For the extraction preparation, 200 mg of wheat leaves was collected. The collected wheat leaves were then transferred to a mortar and ground into a fine powder with liquid nitrogen by using a pestle. Subsequently, an amount of 1.5 mL of high-purity ethanol (99.9%) was added to the sample powdered, and the mixture was allowed to stand on a shaker for 30 min, for thorough mixing. The sample was then centrifuged at a speed of 15,000× *g* for 10 min. The supernatant was collected and stored at −80 °C.

### 4.4. Determination of Total Flavonoid Content (TFC)

The total flavonoid content was determined with the aluminum chloride colorimetric method [58]. Catechin was used to make a standard curve. In a 1.5 mL tube, 100 μL of each sample/standard were mixed with 400 μL of water and 30 μL of sodium nitrite (5%). Afterward, the mixture was incubated at room temperature for 5 min. Then, 30 μL of 10% aluminum chloride was added, and this was allowed to stand for 6 min at room temperature. Next, 200 μL of sodium hydroxide (1 M) and 240 μL of water were then added. After mixing, the sample was transferred to 96-well plate in triplicate. The absorbance value was measured at 515 nm by using a UV spectrophotometer. The results were calculated via interpolation on the standard curve (R^2^ = 0.995). The total flavonoid content was expressed as mg/g of sample.

### 4.5. Determination of Total Polyphenol Content (TPC)

The total polyphenol content was evaluated through a Folin–Ciocalteu assay [59]. Gallic acid was used to make a standard curve. An amount of 50 μL of the standard/sample was mixed with 50 μL of FC reagent and 150 μL of water. The mixture was incubated for 5 min. After mixing, 200 μL of sodium carbonate (10%) and 300 μL of water were added to the solution, and this was allowed to stand for 30 min at room temperature in the dark. The absorbance was measured via spectrophotometry at 750 nm.

### 4.6. Determination of Antioxidant Capacity via DPPH (2,2-Diphenyl-1-picrylhydrazyl) Assay

The scavenging activity was evaluated via a DPPH assay according to earlier studies [60,61], with some minor modifications. Because DPPH is a low-cost assay that relies on the ability to contribute electrons or hydrogen ions, it can be used to measure a sample’s ability to scavenge free radicals in biological systems. For the experiment, 10 mM of DPPH stock solution was prepared in 50 mL of ethanol (99.9%). For the 0.1 mM working solution of DPPH, the stock solution was further diluted in ethanol. Aluminum foil was used to cover the stock and working solution, which were kept at −20 °C in the dark. Gallic acid was used as a positive control. The DPPH radical scavenging of the leaf extracts, and a positive control increased in a dose-dependent manner at concentration range of 2.5, 5, 7.5, 10, and 12.5 μg/mL. Ethanol was set as a negative control. An amount of 450 μL of DPPH working solution was added to 150/150 μL of sample/standard/control. Subsequently, the mixture tubes were covered with aluminum foil and allowed to stand for 30 min at room temperature. The absorbance value was measured at 515 nm using an ELISA reader. The DPPH scavenging activity was calculated using the equation below.
DPPH radical scavenging ability % = (1 − Asample_515_/Acontrol_515_) × 100

### 4.7. Response Surface Methodology (RSM) Design

The response surface methodology used a two-factor and a rotatable central composite design (CCD) consisting of 13 experimental runs created to interpret the optimal concentration of Se acid and Mo oxide treated with HW on the antioxidant property and secondary metabolites of wheat leaves. The three-response values for polyphenol (mg/mL), flavonoid (mg/mL), and DPPH (µg/mL) were analyzed using Minitab 19.2020.1 software (State College, PA, USA). The independent variables and their levels in RSM are shown in Table 3. A response surface plot (3D), a contour plot (2D), and a factorial plot were created to demonstrate the effects of independent variables on the response values. The shape of the contour plot illustrates whether the effect of an independent variable is significant or not. Both the independent factors were significant and had a favorable impact on the experimental response values, as shown by the 3D response surface plot and contour plot.

After obtaining the optimal results using RSM, these results were applied to the growing procedure of wheat seeds again. Three treatments (Mo oxide-HW at 0.06%, Se acid-HW at 0.000005%, and a mixture of both treatments-HW) were selected to obtain the maximal secondary metabolites and antioxidant content. After seven days, the wheat leaves were harvested, and the extract was prepared using conditions identical to those previously used and stored at −20 °C for further experiments.

### 4.8. Metabolites Analysis of Wheat Leaves via LC-MS Chromatography

The metabolites in leaves were analyzed through liquid chromatography (LC/MS) (Ultimate3000, Thermo Scientific, Waltham, MA, USA). Fresh wheat leaves were immediately frozen in liquid nitrogen and grounded into fine powder at −80 °C for further analysis. The metabolites were extracted using a single-phase solvent mixture of chloroform, water, and methanol with a ratio of 2:2:5 *v*/*v*/*v* in an ice bath, centrifuged, and the supernatant was collected and then freeze-dried. Derivatization was carried out by adding 50 μL of methoxyamine hydrochloride in pyridine (20 mg/mL), rapidly vortexing the mixture for 1 min, and then incubating it at 37 °C for 90 min. After adding 80 μL of *N*-methyl-*N*-(trimethylsilyl) trifluoroacetamide (MSTFA) and vortexing for 30 s, the mixture was incubated for 30 min. The liquid chromatography parameters were as follows: column name is Waters Cortex × C18 (2.1 mm × 150 mm, 1.6 μm), and the column temperature is set as 45 °C. The mobile phase A consisted of 0.1% formic acid in water and mobile phase B consisted of 0.1% formic acid in acetonitrile, with a flow rate of 0.35 mL/min. The analysis was performed using the following gradients (Time-A/B): 0 min-99/1, 1 min-99/1, 8 min-95/5, 20 min-70/30, 30 min-0/100, 35 min-0/100, 35.5 min-99/1, and 40 min-99/1. Mass spectroscopy parameters were as follows: electrospray ionization (ESI); MS scan range from 100 to 2000 *m*/*z*. The metabolites were identified by using the NIST 08 library. MetaboAnalyst 3.0 (https://www.metaboanalyst.ca/MetaboAnalyst/, accessed on 27 October 2022) was used to explore the metabolic pathways in wheat leaves with major perturbations.

### 4.9. Cell Culture

The cell line HepG2 was obtained from the American type of culture collection (Rockville, MD, USA). For the cell lining experiments, 10mL of ethanol was added to 100 mg of freeze-dried wheat sample. The mixture was shaken on the shaker for 30 min and then centrifuged at the speed of 15,000 rpm for 10 min. Experiments were conducted with percentage (%) concentration and sample extract defined as 10 mg/mL to 100%. The HepG2 cells were cultured in DMEM supplemented with 10% FBS, 2% penicillin–streptomycin, and 2% HEPES. The HepG2 cells were maintained in a humidified incubator with 5% CO_2_ at a temperature of 37 °C. The culture medium was changed every 2–3 days, and the cells were sub-cultured at approximately 70–80% confluency. The well-grown cells were harvested and seeded into 96-well plates.

### 4.10. Cell Viability Assay

Cell viability was determined by means of an MTT assay, which basically depends on metabolically active cells changing tetrazolium salt (MTT) into a water-insoluble formazan dye. DMSO was used to emulsify the insoluble dye produced during this protocol. The MTT assay was followed by [62]. In our experiment, the cell suspension was counted using a hemocytometer, and then 180 μL of counted cells was dispensed into a 96-well plate at a concentration of 1 × 10^5^ cells/well. The following steps involved cultivating these cells for 24 h at 37 °C in humidified air and 5% CO_2_. After incubation period, the cells were exposed for a further 24 h treated to different concentrations of 10, 20, 25, 50, 75, and 100% of the sample’s treatment. Subsequently, the cells were then given 20 μL of MTT solution, and they were cultured for an additional 4 h to allow for the metabolic conversion of MTT. The produced formazan was dissolved in dimethyl sulfoxide, and the reading was taken at 560 nm through a microplate reader (Biotek, Winooski, VT, USA). A dark blue formazan was produced by the living cells, but no staining was created by the dead cells. The concentration of extract used was not cytotoxic to the HepG2 cells. The control group is set at 100%, and the cell viability of the sample group must be over 80% to determine that there is no toxicity. Cell viability can be measured as follows:Cell viability % = (Sample absorbance/Control absorbance) × 100

### 4.11. OA-Induced Steatosis in HepG2 Cells

The stock solution of oleic acid (OA) was prepared by dissolving 100 mM in DW. This solution of OA was filtered and was then stored at 4 °C before use. For the induction of steatosis, HepG2 cells were initially seeded in 6-well culture plates at a seeding density of 5 × 10^5^ cells/well with a culture media. After reaching a cell density up to 90%, samples were treated with the OA solution for 24 h, and the medium was replaced with serum-free DMEM. Control cells were treated with a serum-free medium throughout the experiment duration.

### 4.12. Oil Red O (ORO) Cell Staining

The morphology and total lipid accumulation of the HepG2 cells were examined using ORO staining. The cell staining experiment was performed according to [57]. First, 500 mg of ORO was mixed in 100 mL of isopropanol mixture and allowed to stand for 20 min to make the stock solution of ORO that was used in this study. The working solution of ORO was prepared at a ratio of 3:2 of ORO stock solution and DW, and the solution was filtered with Whatman filter paper.

The cells were washed with 1 mL of DPBS, and after removing the medium, 2 mL of a formalin solution (10%) was added and the cells were incubated for 30 min. Following the removal of the formalin, the cells were washed twice with 1 mL of DPBS. Subsequently, 2 mL of the ORO working solution was added, and after thoroughly mixing, it was then incubated at room temperature for 30 min.

After removing the ORO solution, the cells were washed 2–5 times with 1 mL of DW. For microscopic observation, 1 mL of DW was added to a 96-well plate. Images of the stained cells were examined with an inverted microscope (Nikon, Tokyo, Japan). Following the microscopic observation, the dyed lipids in the cells were quantified by dissolving them in 2 mL of isopropanol. Afterward, 200 μL of leaching liquor was injected into a 96-well culture plate. The OD value was measured at a wavelength of 510 nm using a microplate reader (Biotek, Winooski, VT, USA).

### 4.13. Determination of Cellular Triglyceride Content in HepG2 Cells

The cellular TG content was measured using a previously described method with some modifications [63,64]. Briefly, HepG2 cells were seeded in 6-well plates (5 × 10^5^ cells/well) and, after the treatment, the cells were washed twice with 1 mL of DPBS. The cells were subsequently collected by centrifugation at 210× *g*. After that, 1 mL of the chloroform/methanol (2:1) solution was dispensed into each well and then transferred to a conical tube. The solution was then centrifuged at 3000 rpm for 10 min, and supernatant was collected. Subsequently, 1% Triton solution was prepared in DPBS. Then, 50 μL of 1% Triton X-100 was added to the nitrogen-concentrated sample. After mixing, it was then transferred to a 1.5 mL EP tube. The cellular triglyceride content was measured by using a TG kit (Asan Pharmacology, Seoul, Republic of Korea). Finally, 200 μL of this solution was added to 96-well plate to undergo OD measurement at a wavelength of 550 nm.

### 4.14. Statistical Analysis

Excel 2013 was used to input and process the experiment’s results, which were displayed as the mean ± SD (Standard Deviation) for each concentration. The experiment included three replicates. The data analysis and graph-making steps were carried out with Minitab version 19.2020.1 software (USA) and Originpro 2022 (64-bit) SR1 v.9.9.0.225. One-way analysis of variance (ANOVA) was used to determine statistical differences between the treatments, and Tukey’s test (significance level; *p* < 0.05) was used for multiple comparisons with a control.

## Figures and Tables

**Figure 1 ijms-24-16742-f001:**
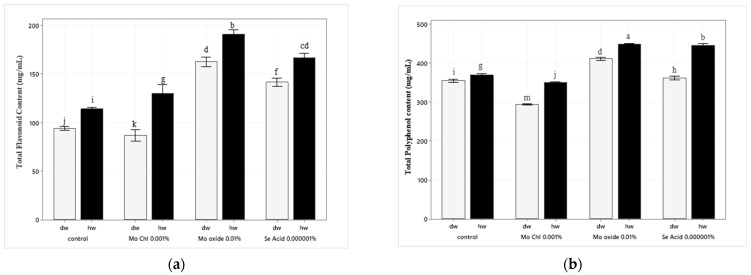
Effects of trace elements treated with HW and DW on secondary metabolite contents of wheat leaves: (**a**) total flavonoid (catechin mg/mL of wheat extract) and (**b**) total polyphenol (GAEmg/mL of wheat extract) content. The data are expressed as the means of three independent experiments. The different letters indicate significant differences among the treatment (ANOVA, followed by Tukey’s test, *α* = 0.05).

**Figure 2 ijms-24-16742-f002:**
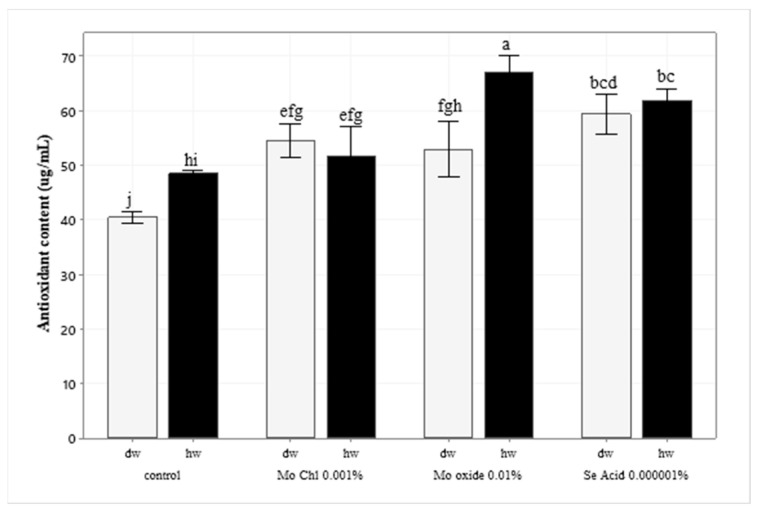
Effect of trace elements treated with HW and DW on the antioxidant capacity of wheat leaves via DPPH (2,2-diphenyl-1-picrylhydrazyl) assay. The data are expressed as the means of three independent experiments. Different letters indicate significant differences among the treatment (ANOVA, followed by Tukey’s test, *α* = 0.05).

**Figure 3 ijms-24-16742-f003:**
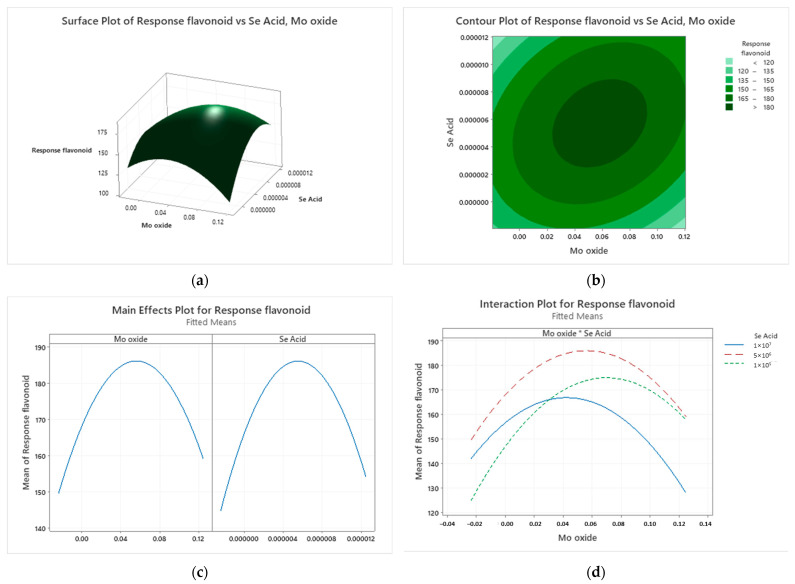
RSM model: (**a**) 2D surface plot, (**b**) 3D contour plot, (**c**) factorial plot, and (**d**) interaction plot (* represents the interaction between Se acid and Mo oxide);the total flavonoid response (mg/mL) affected by Se acid and Mo oxide treated with HW (*α* = 0.05). The blue, green, and red lines represent the interaction at different concentrations of Se acid and Mo oxide. The factorial plot, meanwhile, depicts the optimal concentration ranges of each treatment and interaction plot indicating the significant interaction between trace elements.

**Figure 4 ijms-24-16742-f004:**
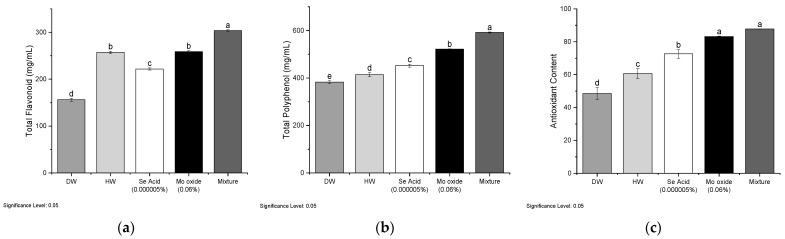
Effect of optimal concentrations on the (**a**) total flavonoid (Catechin mg/mL of wheat extract), (**b**) total polyphenol (GAEmg/mL of wheat extract), and (**c**) antioxidant contents (μg/mL) of wheat leaves. The letters above the bar indicate significant differences among the treatments.

**Figure 5 ijms-24-16742-f005:**
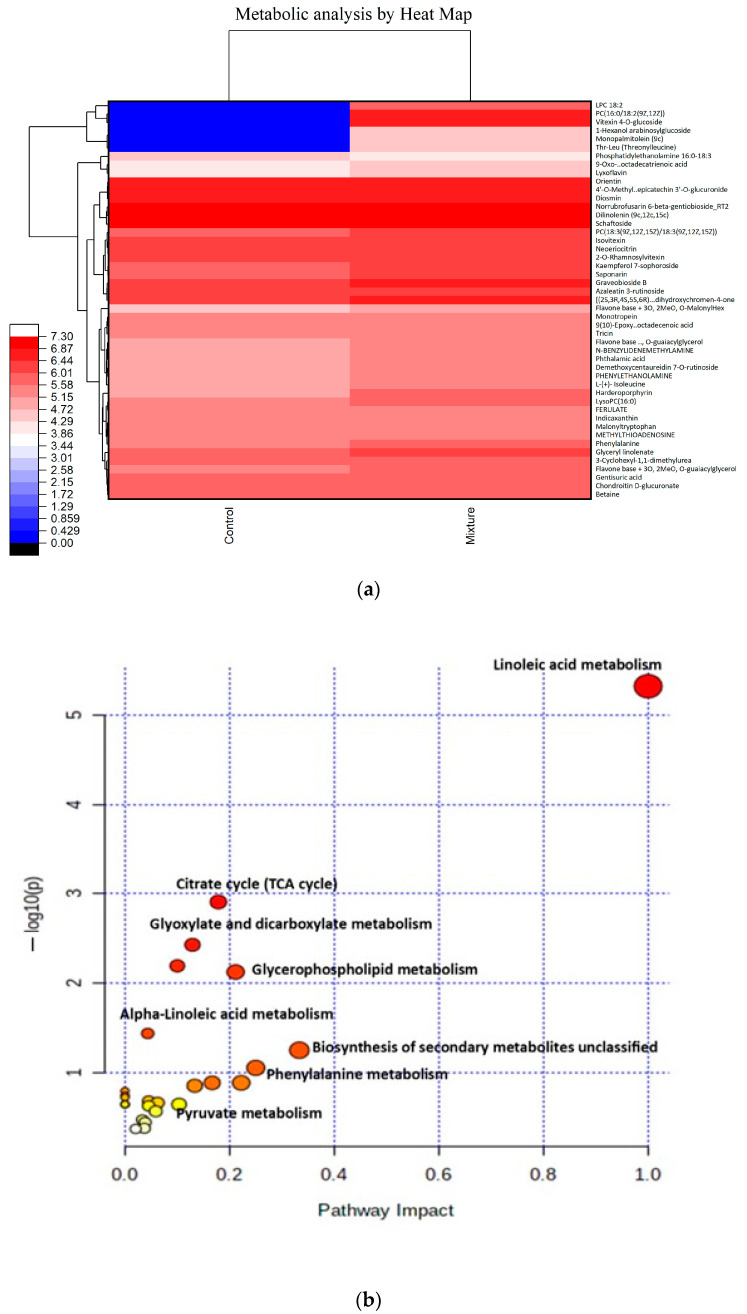
Metabolic profiling reveals the effect of mixture treatment on metabolites of wheat leaves: (**a**) 47 metabolites displayed, both upregulated and downregulated; (**b**) identifying the altered metabolic pathways by the mixture treatment. The color and size of each value based on the *p*-values and pathway impact values.

**Figure 6 ijms-24-16742-f006:**
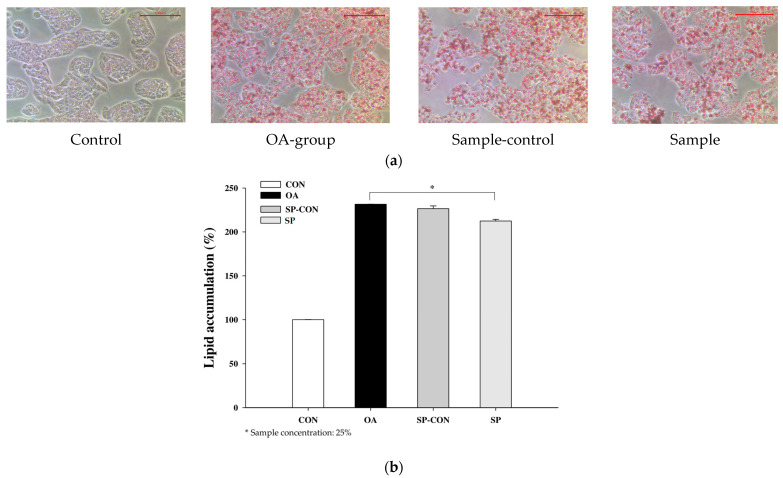
(**a**) After 24 h incubation, the HepG2 hepatocytes cells treated with control and mixture extract, the microscopic visualization of lipid accumulation as stained by oil red O (ORO) in the control group, induction group, control sample (DW), and sample group (mixture treatment). (**b**) Quantitative analysis of lipid accumulation content as stained by oil red O (ORO) in a control group, induction group (OA), SP-CON (DW), and SP (mixture treatment). Data are presented as the mean ± SEM of three separate experiments, (ANOVA; *p* < 0.05).

**Figure 7 ijms-24-16742-f007:**
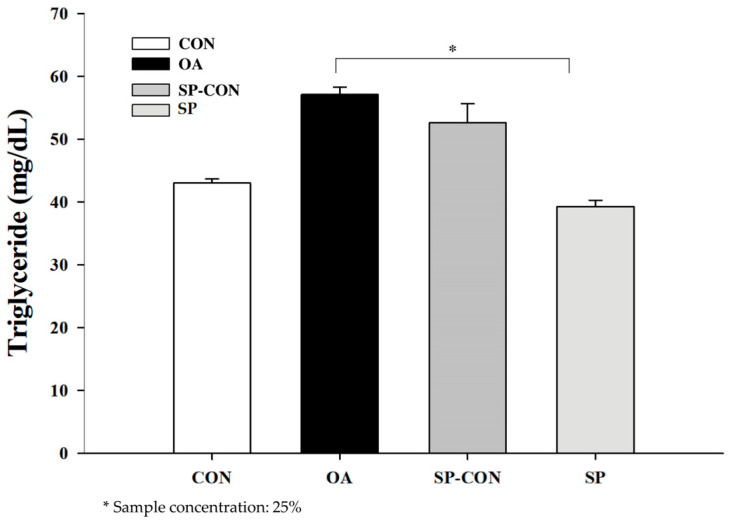
Effect of Mix-HW on cellular triglyceride (TG) accumulation. CON—control group; induction group (OA); SP-CON (DW); and SP (mixture treatment). The concentration of triglycerides was normalized with protein content. Data are presented as the mean ± SEM of three independent experiments (ANOVA; *p* < 0.05).

**Table 1 ijms-24-16742-t001:** Biomass factors (i.e., area, length, and weight) of wheat leaves following application of non-optimal concentrations of trace elements treated with HW and DW.

Treatments	Area (cm^2^)	Length (cm)	Weight (g)
Control (DW)	9.095 ± 1.62 ^efg^	13.839 ± 0.858 ^bc^	0.126 ± 0.004 ^de^
Hydrogen water (HW)	14.392 ± 0.48 ^abc^	15.873 ± 0.522 ^ab^	0.138 ± 0.005 ^b^
Mo oxide DW (0.01%)	12.603 ± 0.85 ^bcd^	16.148 ± 0.356 ^abc^	0.136 ± 0.008 ^bc^
Mo oxide HW (0.01%)	17.398 ± 0.68 ^a^	17.065 ± 0.740 ^a^	0.148 ± 0.002 ^a^
Mo Chl-DW (0.001%)	14.64 1 ± 0.240 ^ab^	15.813 ± 1.353 ^abc^	0.140 ± 0.003 ^b^
Mo Chl-HW (0.001%)	11.545 ± 1.460 ^cde^	15.053 ± 0.756 ^abc^	0.140 ± 0.001 ^b^
Se Acid-DW (0.000001%)	9.917 ± 0.839 ^def^	13.7888 ± 0.661 ^bc^	0.122 ± 0.002 ^ef^
Se Acid-HW (0.000001%)	16.041 ± 0.600 ^a^	16.111 ± 0.792 ^abc^	0.148 ± 0.0004 ^a^

Different letters indicate significant differences among treatments (ANOVA, followed by Tukey’s test, *α =* 0.01).

**Table 2 ijms-24-16742-t002:** The corresponding response values (experimental) optimized with central composite design (CCD).

Run	Flavonoid (mg/mL)	Polyphenol (mg/mL)	Antioxidant Content (µg/mL)
1	143.253	340.252	40.1379
2	140.222	338.526	40.9205
3	140.256	337.353	40.3808
4	168.524	399.985	48.7736
5	165.326	405.145	48.4558
6	165.555	410.526	48.3178
7	161.071	410.121	55.2924
8	161.071	410.121	51.8741
9	164.357	412.667	51.6942
10	188.357	449.667	68.3358
11	191.786	449.555	65.9070
12	191.071	448.970	66.7166
13	193.929	420.222	62.5787

**Table 3 ijms-24-16742-t003:** The independent variables and their levels in response surface methodology.

Factors	Name	Type	Low Actual	High Actual	Low Coded	High Coded
X_1_	Mo oxide	Numeric	1 × 10^−3^	0.100	−1.000	1.000
X_2_	Se acid	Numeric	1 × 10^−7^	1 × 10^−5^	−1.000	1.000

## Data Availability

Data is contained within the article and Appendix A.

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
