# Peer review of "Enhancement of Growth and Secondary Metabolites by the Combined Treatment of Trace Elements and Hydrogen Water in Wheat Sprouts"

_ijms, 2023, doi:10.3390/ijms242316742_

Round 1
Reviewer 1 Report
Comments and Suggestions for Authors
Dear Authors,
Reviewer comments ijms-2708581
The manuscript entitled „Enhancement of growth and secondary metabolites by the combined treatment of trace elements and hydrogen water in wheat sprouts“ studies the effects of hydrogen water, i.e., water treated with photocatalytic water splitting containing hydrogen molecules, together with Mo(VI)-oxide and Se-acid (selenous acid) on wheat leaf secondary metabolites content, namely total flavonoid and total polyphenol contents. In addition, the study shows that the mixture of Se-acid and Mo-oxide reduces lipid accumulation in HepG2 hepatocytes.
It seems that wheat leaves secondary metabolites content and HepG2 hepatocytes lipid contents are two systems on which the effects of hydrogen water and Mo-oxide and Se-acid (selenous acid) were studied. Is it true? Or should the therapeutical effects of wheat leaves extracts be studied in hepatocytes?
I have several comments on the present manuscript:
In Materials and methods, line 364, the wheat cultivar or genotype used for the experiments has to be given. Not just wheat (T. aestivum) seeds, but which kind of wheat genotype has to be given!
In Materials and methods, the apparatus used for photocatalytic water splitting and hydrogen water preparation has to be given and the basic information on the conditions used for hydrogen water preparation have to be described! The content of hydrogen in hydrogen water developed by photocatalytic water splitting has to be quantified. Some quantification has to be provided.
In Materials and methods, Statistical analysis, the 0.05 significance level has to be specified as a threshold for Tukey´s test.
In Results, Figure 5a, an appropriate scale bar has to be added to the photos of hepatocytes. Moreover, it has to be given in the figure legend that the sample means HepG2 hepatocytes!
Terminology:
Line 276: Use the term „low-molecular weight“, not „low-molecular mass“.
Line 353: Write „Mo(VI) oxide“ , not „Mo (Vi) oxide“.
Line 247: The expression „The geological accumulation of lipid droplets“ has to be explained!! What should it mean?
Line 341: The expression „finishing pigs“ has to be explained!
Formal comments on the text:
Introduction, line 92: Add the words „experiments indicating“ following the words „This study represents…“ and replace „then“ with „than“ in the statement: „This study represents experiments indicating that combining various antioxidants strategies will be more beneficial than only enhancing the antioxidant capacity by a single treatment.“
Results, line 190: Modify the 2.5. heading as follows: „2.5. Wheat cultivation, extraction, and determination of secondary metabolites under optimal conditions“.
Table 1 legend. I absolutely do not understand Table 1 legend text: „Represents the independent variables and their levels in response surface methodology“ – the Table 1 shows levels of total flavonoid, total polyphenol, and antioxidant content in a series of 1-13 runs?? I think that Table 1 legend has to be modified to correspond to the Table 1 content.
Line 195: Add the word „respectively“ following the statement: „Mo oxide and Se acid under the optimal concentration significantly increased the total flavonoid content by 22% and 12.5%, respectively, as compared to the non-optimal concentration…“
Line 260: Remove the words „accumulation of“ in the statement: „To determine triglyceride (TG) accumulation,….“
Line 285: Replace the word „improve“ with „increase“ in the statement. „This study looked at several circumstances to increase wheat leaf secondary metabolites content…“
Results, lines 142, Figure 2 legend, line 150, Line 403 in Materials and methods, 4.6. heading, the full name for „DPPH assay“ has to be given in the heading!
Line 435: Add a comma following the words „After seven days,…“
Line 450: Add a space between the number and the corresponding unit in „30 s“.
Lines 479, 480: Add a space between the number and the corresponding unit in „24 h“ (twice) and „37 °C“.
Line 491: Add a space between the number and the corresponding unit in „100 mM in DW.“
Line 520: Add a comma between the words „and“ and „after the treatment,…“
Final recommendation: Reconsider after a major revision.

Comments on the Quality of English LanguageDear Authors,
Reviewer comments ijms-2708581
The manuscript entitled „Enhancement of growth and secondary metabolites by the combined treatment of trace elements and hydrogen water in wheat sprouts“ studies the effects of hydrogen water, i.e., water treated with photocatalytic water splitting containing hydrogen molecules, together with Mo(VI)-oxide and Se-acid (selenous acid) on wheat leaf secondary metabolites content, namely total flavonoid and total polyphenol contents. In addition, the study shows that the mixture of Se-acid and Mo-oxide reduces lipid accumulation in HepG2 hepatocytes.
It seems that wheat leaves secondary metabolites content and HepG2 hepatocytes lipid contents are two systems on which the effects of hydrogen water and Mo-oxide and Se-acid (selenous acid) were studied. Is it true? Or should the therapeutical effects of wheat leaves extracts be studied in hepatocytes?
I have several comments on the present manuscript:
In Materials and methods, line 364, the wheat cultivar or genotype used for the experiments has to be given. Not just wheat (T. aestivum) seeds, but which kind of wheat genotype has to be given!
In Materials and methods, the apparatus used for photocatalytic water splitting and hydrogen water preparation has to be given and the basic information on the conditions used for hydrogen water preparation have to be described! The content of hydrogen in hydrogen water developed by photocatalytic water splitting has to be quantified. Some quantification has to be provided.
In Materials and methods, Statistical analysis, the 0.05 significance level has to be specified as a threshold for Tukey´s test.
In Results, Figure 5a, an appropriate scale bar has to be added to the photos of hepatocytes. Moreover, it has to be given in the figure legend that the sample means HepG2 hepatocytes!
Terminology:
Line 276: Use the term „low-molecular weight“, not „low-molecular mass“.
Line 353: Write „Mo(VI) oxide“ , not „Mo (Vi) oxide“.
Line 247: The expression „The geological accumulation of lipid droplets“ has to be explained!! What should it mean?
Line 341: The expression „finishing pigs“ has to be explained!
Formal comments on the text:
Introduction, line 92: Add the words „experiments indicating“ following the words „This study represents…“ and replace „then“ with „than“ in the statement: „This study represents experiments indicating that combining various antioxidants strategies will be more beneficial than only enhancing the antioxidant capacity by a single treatment.“
Results, line 190: Modify the 2.5. heading as follows: „2.5. Wheat cultivation, extraction, and determination of secondary metabolites under optimal conditions“.
Table 1 legend. I absolutely do not understand Table 1 legend text: „Represents the independent variables and their levels in response surface methodology“ – the Table 1 shows levels of total flavonoid, total polyphenol, and antioxidant content in a series of 1-13 runs?? I think that Table 1 legend has to be modified to correspond to the Table 1 content.
Line 195: Add the word „respectively“ following the statement: „Mo oxide and Se acid under the optimal concentration significantly increased the total flavonoid content by 22% and 12.5%, respectively, as compared to the non-optimal concentration…“
Line 260: Remove the words „accumulation of“ in the statement: „To determine triglyceride (TG) accumulation,….“
Line 285: Replace the word „improve“ with „increase“ in the statement. „This study looked at several circumstances to increase wheat leaf secondary metabolites content…“
Results, lines 142, Figure 2 legend, line 150, Line 403 in Materials and methods, 4.6. heading, the full name for „DPPH assay“ has to be given in the heading!
Line 435: Add a comma following the words „After seven days,…“
Line 450: Add a space between the number and the corresponding unit in „30 s“.
Lines 479, 480: Add a space between the number and the corresponding unit in „24 h“ (twice) and „37 °C“.
Line 491: Add a space between the number and the corresponding unit in „100 mM in DW.“
Line 520: Add a comma between the words „and“ and „after the treatment,…“
Final recommendation: Reconsider after a major revision.
Author Response
Comments from Reviewer 1.
- Comment 1: It seems that wheat leaves secondary metabolites content, and HepG2 hepatocytes lipid contents are two systems on which the effects of hydrogen water and Mo-oxide and Se-acid (selenous acid) were studied. Is it true? Or should the therapeutical effects of wheat leaves extracts be studied in hepatocytes?
Response: Respected reviewer, it is worth noting that this study design appears to be aimed at determining the influence of the combined treatments (HW, Mo-oxide, and Se-acid) on wheat plants. Moreover, wheat secondary metabolites and growth indices were examined to determine how these treatments affected the biochemical profile and growth. The focus of this study was to evaluate how these treatments affected wheat plants and indirectly explore their potential therapeutic impact through observed changes in metabolic profile of wheat sprouts. From the metabolic profile data, we observed some of the increased metabolites are linked to the inhibition of lipid accumulation. So, indirectly we infer the therapeutic effect through observed metabolic changes rather than directly studying the effect of wheat leaf extract on hepatocytes.
- Comment 2: In Materials and methods, line 364, the wheat cultivar or genotype used for the experiments has to be given. Not just wheat (T. aestivum) seeds, but which kind of wheat genotype has to be given!
- Response: Thank you for pointing out this, we revised the manuscript by providing the information of cultivars in the methodology. Line no,396.
- Comment 3: In Materials and methods, the apparatus used for photocatalytic water splitting and hydrogen water preparation has to be given and the basic information on the conditions used for hydrogen water preparation have to be described! The content of hydrogen in hydrogen water developed by photocatalytic water splitting has to be quantified. Some quantification has to be provided.
Response: Thank you, for raising that point in our manuscript. We added this information to our manuscript. Basically, we provided information on how we prepared the hydrogen water in the laboratory. Line no 398-404.
- Comment 4: In Materials and methods, Statistical analysis, the 0.05 significance level has to be specified as a threshold for Tukey´s test.
Response: Agree, we revised the manuscript accordingly. Line no 579.
Comment 5: In Results, Figure 5a, an appropriate scale bar has to be added to the photos of hepatocytes. Moreover, it has to be given in the figure legend that the sample means HepG2 hepatocytes!
Response: As per your suggestion, we provided the scale bar on figure 5a, as well mentioned the information of HepG2 hepatocytes in the figure legend. Line no 275.
Comment 6: Line 276: Use the term „low-molecular weight“, not „low-molecular mass“.
Response: We revised the manuscript according to your comment. Line no 298.
Comment 7: Line 353: Write „Mo(VI) oxide“ , not „Mo (Vi) oxide“.
Response: We revised the manuscript according to your comment. Line no 385.
Comment 8: The expression „The geological accumulation of lipid droplets “has to be explained!! What should it mean?
Response: Thank you for pointing this, we removed this information, as the accumulation of lipid droplets can also be used.
Comment 9: Line 341: The expression „finishing pigs“ has to be explained!.
Response: We agree to this, and we have explained about the finishing pigs in our manuscript. Basically, finishing pigs refers to the pigs that have completed their growth cycle and are ready for market. Line No 371-372.
Comment 10: Introduction, line 92: Add the words „experiments indicating“ following the words „This study represents…“ and replace „then“ with „than“ in the statement: „This study represents experiments indicating that combining various antioxidants strategies will be more beneficial than only enhancing the antioxidant capacity by a single treatment.“
Response: We revised the manuscript according to your suggestion. Line No 94-95
Comment 11: Results, line 190: Modify the 2.5. heading as follows: „2.5. Wheat cultivation, extraction, and determination of secondary metabolites under optimal conditions “.
Response: We revised the heading 2.5 according to your suggestion. Line No 209
Comment 12: Table 1 legend. I absolutely do not understand Table 1 legend text: „Represents the independent variables and their levels in response surface methodology“– the Table 1 shows levels of total flavonoid, total polyphenol, and antioxidant content in a series of 1-13 runs?? I think that Table 1 legend has to be modified to correspond to the Table 1 content.
Response: We revised table 1 legend according to your suggestion. Line No 171-172
Comment 13: Line 195: Add the word „respectively “following the statement: „Mo oxide and Se acid under the optimal concentration significantly increased the total flavonoid content by 22% and 12.5%, respectively, as compared to the non-optimal concentration…“.
Response: We revised the manuscript according to your suggestion. Line No 214
Comment 14: Line 260: Remove the words „accumulation of“ in the statement: „To determine triglyceride (TG) accumulation,….“
Response: We revised the manuscript according to your suggestion. Line No 283
Comment 15: Line 285: Replace the word „improve“ with „increase“ in the statement. „This study looked at several circumstances to increase wheat leaf secondary metabolites content…“
Response: We revised the manuscript according to your suggestion. Line No 308
Comment 16: Results, lines 142, Figure 2 legend, line 150, Line 403 in Materials and methods, 4.6. heading, the full name for „DPPH assay“ has to be given in the heading!
Response: We provide the full name of DPPH as your suggestion in the manuscript. Results, lines 152, Figure 2 legend, line 162, Line 338 in Materials and methods, 4.6. heading line, no 446.
Comment 17: Line 435: Add a comma following the words „After seven days,…“
Response: We revised the manuscript according to your suggestion. Line No 479
Comment 18: Line 450: Add a space between the number and the corresponding unit in „30 s“.
Response: We revised the manuscript according to your suggestion. Line No 513
Comment 19: Lines 479, 480: Add a space between the number and the corresponding unit in „24 h“ (twice) and „37 °C“
Response: We revised the manuscript according to your suggestion. Line No 523
Comment 20: Line 491: Add a space between the number and the corresponding unit in „100 mM in DW. “
Response: We revised the manuscript according to your suggestion. Line No 535
Comment 21: Line 520: Add a comma between the words „and“ and „after the treatment,…“
Response: We revised the manuscript according to your suggestion. Line No 564
Reviewer 2 Report
Comments and Suggestions for Authors
The submitted manuscript is important and topical from a practical point of view. It is a topic that has the potential for further citation. The manuscript is relatively carefully written. The abstract captures the main thesis and research findings. The results are rather descriptive, unfortunately there are no measured values, so this would be useful to add. The overview gives a comprehensive picture of the subject. It might be useful to add the objectives of the research. The methodology should be supplemented with wheat variety. It would also be useful to add the growing conditions and the developmental stage of the plant. Again, the results are rather descriptive without giving the measured values. This should be added. It would be useful to enlarge and improve the quality of the graphs. The discussion is descriptive. I recommend to focus again on the evaluation of the actual results and their discussion. The authors mainly cite older literature. Please update them. Are all sources necessary?
Author Response
Comments from Reviewer 2.
Comment 1: The results are rather descriptive, unfortunately there are no measured values, so this would be useful to add. It might be useful to add the objectives of the research. The methodology should be supplemented with wheat variety. It would also be useful to add the growing conditions and the developmental stage of the plant. The discussion is descriptive. I recommend focussing again on the evaluation of the actual results and their discussion. The authors mainly cite older literature. Please update them. Are all sources necessary?
Response: Agree, we revised the results according to your valuable suggestion. We tried to provide the more measured values in the result section. Moreover. We provided the objective of our study in Line No 96. Thank you for pointing out the point related to wheat variety, we provided the wheat variety as well as we tried to provide the detail methodology section. We tried to provide the details of hydrogen water preparation, as well as how we grow wheat sprouts Line No 396-412. All the graphs were modified according to your suggestion. Moreover, we tried to support our discussion with the previous studies, that is why we tried to add the other research papers. But in accordance with your suggestion, we tried to revise our discussion section. Moreover, we also provided additional latest research paper Reference 12. Thank you once again on all the valuable comments, we tried our best to revise this manuscript according to your suggestion.
Round 2
Reviewer 1 Report
Comments and Suggestions for Authors
Dear Authors,
I have no further comments on the revised manuscript. I can recommend the revised manuscript for publication in IJMS.
Final recommendation. Accept.
Reviewer 2 Report
Comments and Suggestions for Authors
The submitted revised manuscript was corrected and modified according to the requirements of the individual reviewers. Changes have been noted in the text. The authors have accepted the criticisms and comments. After comparing the original version with the present version, I must conclude that the manuscript is qualitatively suitable for publication. In its present form, the manuscript also has potential for citation.